# Trans* Pregnancy and Lactation: A Literature Review from a Nursing Perspective

**DOI:** 10.3390/ijerph17010044

**Published:** 2019-12-19

**Authors:** Jesús Manuel García-Acosta, Rosa María San Juan-Valdivia, Alfredo David Fernández-Martínez, Nieves Doria Lorenzo-Rocha, Maria Elisa Castro-Peraza

**Affiliations:** 1Faculty of Nursing, University of La Laguna, Canary Islands Public Health Service, 38010 Tenerife, Canary Islands, Spain; 2Madrid Health Service (SERMAS), Hospital Central de la Defensa, 28047 Madrid, Spain; rosecarina32@yahoo.es; 3Department of Education Counseling Universities, Culture and Sports, Government of the Canary Islands, 38010 Tenerife, Canary Islands, Spain; alfdavfermar@gmail.com; 4The Faculty of Nursing Nª Sª de La Candelaria, University of La Laguna, Canary Islands Public Health Service, 38010 Tenerife, Canary Islands, Spain; extnlorenzo@ull.edu.es (N.D.L.-R.); mcastrop@ull.edu.es (M.E.C.-P.)

**Keywords:** nursing, nursing care, perinatal care, transgender individuals, gender dysphoria, maternal lactation, pregnancy, lactation

## Abstract

Pregnancy and lactation involve two aspects that are socially and culturally associated with women. However, there are a few biological differences between male and female breast tissue. Lactation and pregnancy are viable processes that do not depend on sex. Even for the latter, it is only necessary to have an organ capable of gestation. Ways to favor mammogenesis and lactogenesis in trans* women have been established. There are protocols to promote lactation in trans* women, usually used for adoptive mothers or those whose children have been born through gestational surrogacy. Chestfeeding a baby could be the cause of feelings as diverse as gender dysphoria in the case of trans* men, and euphoria and affirmation of femininity in trans* women. This study involves a review of the available scientific literature addressing medical aspects related to pregnancy and lactation in trans* individuals, giving special attention to nursing care during perinatal care. There are scarce studies addressing care and specifically nursing care in trans* pregnancy and lactation. Our study indicates the factors that can be modified and the recommendations for optimizing the care provided to these individuals in order to promote and maintain the lactation period in search of improvement and satisfaction with the whole process.

## 1. Introduction

A transgender, or trans*, person is someone who does not feel they identify with their sex assigned at birth (the * symbol is meant to be inclusive of all identities/genders). Trans* people experience their gender as different to that normatively expected of their assigned sex. This identity is independent of genotype, sexual orientation, and behavior [1,2,3]. This discrepancy can potentially be a source of gender dysphoria, and experiences of distress or anxiety with respect to the gender and the physical body [4]. To date, no epidemiological studies have been conducted reporting the incidence and prevalence of trans* identities [5]. Studies, such as that conducted by Winter et al., have estimated that there are between eight and twenty-five million people worldwide who identify themselves as trans* individuals [6]. In this sense, the American Psychiatric Association (APA), in its fifth edition of the Diagnostic and Statistical Manual of Mental Illnesses (DSM), establishes a prevalence figure of trans* individuals ranging from 0.003% in trans* women up to 0.014% in trans* men [7]. However, these figures seem to be underestimated, given that not all trans* individuals request healthcare, hormonal therapies, or undergo sexual reassignment surgeries.

The transition of gender is a personal, variable, fluid, and dynamic process of the experience of gender identity, in search of alignment with the gender identity that the person feels. In the case of resorting to hormonal treatment, it will be an exogenous testosterone intake for trans* men [1,8]; whereas, for trans* women, estrogen is usually combined as a feminizing hormone along with medications commonly used to reduce the effects of androgenization such as spironolactone [1,5], cyproterone acetate, and gonadotropin-releasing hormone (GnRH) agonists [5]. This process reduces both the levels of endogenous testosterone and the activity of testosterone in the tissues, thus obtaining a noticeable decrease in trans* male characteristics, such as, for example, body hair [5]. The identification of important aspects in this transition helps health professionals make decisions and improves care provided to trans* patients [9].

Some trans* men who keep their uterus are able to become pregnant and give birth [8,10,11,12]. Some of them decide to chestfeed their babies and require specialized support to do so [4,13]. Although organizations and health professionals have become more aware of reproductive health and lactation in trans* individuals in recent years [14], there is a scarce number of studies addressing this topic [14,15,16,17,18], as well as scarce amount of scientific literature that describes the experiences of pregnancy in trans* men [3,4,10,18,19,20,21,22], especially insufficient in the field of nursing [16].

Unlike trans* men, trans* women do not have a reproductive system that allows for gestation and, therefore, cannot become pregnant. However, they are capable of developing breast tissue that is histologically and radiologically indistinguishable from that of cisgender women [1]. Thus, they can also chestfeed effectively [23,24]. As assisted reproduction techniques advance, cisgender women are no longer the only ones involved in pregnancy, childbirth, and lactation [17].

Although both men and women have breast tissue, the word “breast” is more associated with the female sex, which will generate discomfort in the trans* men who feel more comfortable with the term “chestfeeding” than with the term “nursing” [13]. On the other hand, for trans* women, breast development is an important marker of physical feminization [1,2].

Even though trans* individuals have experienced significant progress in their social acceptance, stigma and discrimination persist [10], including in health services [3,25,26]. In this regard, nurses need more training on how to support trans* patients during pregnancy and lactation [17]. There is an important gap between what is taught in professional schools, what is taught in graduate programs at university, and the real needs of trans* individuals [10,27].

Therefore, the aim of the present study is to explore the existing scientific literature addressing the lactation and pregnancy processes in trans* individuals and the recommendations for perinatal care.

## 2. Materials and Methods

The present study is based on a review of the available scientific literature, aimed at identifying previous studies addressing the topic under consideration. The articles found in this search were obtained from the following databases: Virtual Health Library (VHL); LILACS; Cuiden; SciELO; PubMed; Web of Science (WOS); CINAHL; JBI; and MEDLINE.

The following keywords or medical subject headings descriptors (MeSH) were used for the search: Nursing; Nursing care; Perinatal care; Transgender individuals; Lactation; Breastfeeding; Pregnancy; and Gender dysphoria, using Boolean operators “AND” and “OR” to increase the coverage of studies. We included both quantitative and qualitative studies without discriminating publication dates or languages. Additionally, we performed an inverse search based on the studies found.

A total of 212 studies with the main objective of trans* pregnancy and breastfeeding were identified; after a critical reading of the documents, we included a total of 53 scientific studies addressing medical aspects related to pregnancy and lactation in trans* individuals that were obtained from articles, books, and online resources, giving special attention to nursing care during perinatal care in this review. Abstracts were reviewed for relevance, and relevant manuscripts were reviewed in full. Discussion sessions were held to increase the consensus of the group while screening and analyzing. During the consensus meetings, themes were identified through observation and discussion. The validity and reliability of the selection were given by the degree of evidence, the recommendations of the references, and the applicability to our study.

The complete identification and selection process is shown in a PRISMA diagram (Figure 1).

## 3. Results

The main results of this search are included in the following sections which will be treated below: gestation and lactation in trans* men, the breastfeeding process in trans* women, barriers in healthcare, and recommendations on breastfeeding and chestfeeding.

Refer to Figure 2 to see an overview of the articles and the theme that they align with.

### 3.1. Gestation and Chestfeeding in Trans* Men

Some of the individuals who need obstetric care are not ciswomen [8]. In those trans* men who have undergone surgeries during their transition, such as hysterectomy, metaoidioplasty, or phalloplasty, pregnancy is not a possibility [21]. 

Trevor MacDonald found that the majority of trans* men choose to undergo a surgical process of chest masculinization, which differs from a conventional mastectomy or breast reduction in that the goal is to create a male breast, maintaining part of the mammary gland [13], which will also allow them to chestfeed in the future if they wish. In this sense, the “periareolar” approach, in which the nipples remain intact, seems to show better outcomes in future lactations, unlike the “double incision” approach, which includes nipple grafts, reduces sensitivity, and does not always keep the milk ducts intact [13].

Not all trans* men who have give birth want to chestfeed. Sometimes, this fact results from mental health issues and feelings of dysphoria [7]. It is always a personal decision. Others, however, wish to chestfeed, choosing to avoid chest masculinization surgery in order to be able to produce enough milk [13].

Regarding hormone treatment, testosterone is the key hormone in masculinization therapy. Testosterone can be administered by intramuscular injections, transdermal patches, topical gels, or implants [28,29]. Hormone therapy with testosterone will cause a series of consequences, such as: amenorrhea, cessation of ovulation, and the appearance of typically cismale secondary characteristics such as low-pitched voice, facial hair growth, and the pattern of androgenic baldness [13].

Hormone therapy should be discontinued if gestation is desired in order to recover ovulation cycles, which takes between eight and twelve months to resume after testosterone withdrawal [30,31,32,33]. If pregnancy is achieved, testosterone treatment should be abandoned, given that it has teratogenic effects on the fetus [5,10,13,34], and it is safe to conceive a few months after cessation given its high metabolic rate [13]. This interruption of hormonalization during pregnancy will reverse the main changes already established, such as: increased breast tissue, redistribute fat in the hips, reduce facial hair [4], and decrease bone density [5]. It also causes intense mood swings such as increased gender dysphoria [18,21]. It can have a great damaging impact [21], especially in men who have not undergone chest masculinization due to the development of breast tissue [4,13] and feelings of anxiety, depression, isolation, and loneliness [10,13,21,22].

A strategy for the management of dysphoria generated by breast augmentation involves the use of a bandage or a compressive elastic garment to flatten the breasts, which is known as a “chest binder” or “binder.” However, its use can cause glandular tissue involvement of the breast [13]. In addition to the binder, many trans* men resort to coping strategies in view of the visibility of their pregnancy, such as: impersonating cisgender women; going unnoticed as an obese cisgender man; or becoming visible as a trans* pregnant man [8,11,35].

The results obtained also indicated that there was a higher proportion of caesarean sections by choice [18], mainly in those trans* men who had previously taken hormones and considered vaginal delivery as a disturbing experience [22]. This fact poses a challenge for specialized obstetric care, given that there is a significant lack of knowledge about the perinatal approach. [10,21,34].

After giving birth by vaginal delivery or caesarean section, the restoration of hormonal therapy with testosterone can interfere with the hormones necessary for the production of milk [13], such as prolactin, insulin, and hydrocortisone, although the use of testosterone seems to be safe because it is not significantly excreted through milk and does not have an effect on the newborn [10]. Many trans* men do not want to chestfeed because they recognize that chestfeeding is a turning point. They describe it as an anguishing experience and even claim to be the pinnacle of gender dysphoria [21], which leads them to suppress chestfeeding [10]. Others, however, link chestfeeding to a natural form of attachment and strengthening of the bond with their babies [4].

### 3.2. The Breastfeeding Process in Trans* Women

Sonnenblick et al. found that approximately 60% of trans* women resorted to breast augmentation regardless of whether they were receiving hormonal treatment or not. An essential element in the transition of trans* women is the development of breast tissue [2], which is an important marker of physical feminization [1,2].

Whereas testosterone is administered in trans* men, estrogen is the leading hormone in the case of trans* women [1,2]. After the start of hormonal therapy, there is an initial development between the first three to six months starting with a small subareolar breast buds, followed by increased breast tissue development and increased volume [2]. Breast development is not comparable to that of cisgender women, maintaining an immature chest and smaller breasts [36]. The maximum growth will not be achieved until the second year [2], showing no relationship with doses and type of treatments used [37]. In addition, estrogenic therapy is usually combined with medications commonly called “anti-androgens,” which are used to reduce the effects of testosterone, such as: spironolactone [1,5,28,29], cyproterone acetate, and GnRH [5].

The breast tissue that develops, using the standard estrogen hormone, is radiographically [2] and histologically [38] indistinguishable from that of any cisgender woman. Tissue changes derived from a therapy with high estrogen levels during the transition, unlike what happens in gynecomastia, leads to the development of galactophores ducts, lobes, and alveoli. This way, the glandular volume increases, which also turn out to be identical to that of cisgender women [2].

The ability to induce non-puerperal functional breastfeeding has been previously documented. Most protocols are based on the protocol created by Jack Newman, a pediatrician and founder of the Newman Breastfeeding Clinic & Institute [39,40], so that Lenore Goldfarb, a ciswoman, was able to breastfeed her son born by gestational surrogacy. This protocol is included in his book titled “Newman-Goldfarb” [41]. The text is based on the use of certain medications and breast stimulation. The expected outcome is to imitate the physiological development of the mammary gland during pregnancy, the progressive increase in serum prolactin levels after childbirth, and the stimulation and extraction of milk [42,43]. To that end, estrogenic therapy is complemented with progesterone and is responsible for the increase in duct branching and maturation [1]. The following basic guideline for inducing non-puerperal breastfeeding has been reported: first, estradiol and progesterone are increased in a way that reproduce the high levels of pregnancy; then, a galactagogue, such as domperidone, is used to increase prolactin levels, together with stimulation produced by a breast pump. In parallel, there is secretion of prolactin, which plays a key role in mammogenesis, and oxytocin, which will favor the ejection of milk. Finally, the levels of estradiol and progesterone are reduced by mimicking the natural postpartum process [23,44,45].

The United States government agency responsible for the regulation of food, medicines, cosmetics, medical devices, biological products, and blood products (Food and Drug Administration—FDA), considers the current use of domperidone an effective galactagogue, posing unknown risks to infants [46]. In this sense, the study conducted by Reisman and Goldstein in 2018 indicated the effectiveness of domperidone in a trans* woman in achieving milk secretion together with the use of a breast pump [23].

### 3.3. Barriers in Healthcare

Although the equity of the trans* community has been gaining strength worldwide, there is still a lack of education and necessary support regarding the lactation process. This fact might be related to the lack of experiences and knowledge of this event in the trans* population [47]. Recent studies have indicated the lack of works addressing this topic in the field of nursing compared to other health professions. There is a minimum number of scientific studies related to the role of nurses in care provided to trans* individuals, especially during perinatal care [17].

Gaining access to healthcare, in general terms, can be a real challenge, especially for trans* men, due to discrimination, rejection of treatment, and lack of knowledge and cultural understanding on the part of health professionals [5,48,49]. In this sense, there are investigations that have pointed out stigmatization, violence, oppression, and discrimination against this population that faces unique and specific barriers when receiving healthcare [5,17], such as misinformation about the short- and long-term effects of testosterone in the reproductive organs, the ease of conception, pregnancy, mental health, and the lactation process [8].

Regarding the barriers to healthcare perceived by trans* individuals, a study conducted by Grant et al. revealed that 19% of trans* individuals had been denied healthcare because of their gender identity, 50% had to teach their health providers about trans* health issues, and 28% had delayed the search for healthcare due to fear of being discriminated against [49].

Health professionals have observed barriers in providing healthcare, which were mainly derived from a general shortage of knowledge about trans* health and guidelines for trans* patient care [26,50,51,52]. Many have also reported lack of preparation and concern about how to address the transition process of their patients [52].

The health services were not familiar enough with trans* identities and exhibited problems when interacting with trans* patients, which made it difficult to support the transition, especially in the pregnancy of trans* men. This way, this fact demonstrates the need to create inclusive and specialized environments [21].

Three areas of special needs for the change have been established, namely: (a) development of equitable health systems; (b) training of health professionals; and (c) training based on scientific evidence [47]. In this sense, a recent study conducted by Duckett and Rudd indicated the importance of using an inclusive language, establishing guidelines for health professionals with little knowledge and experiences relating to individuals from the trans* population [53]. All health professionals—i.e., administrators, nurses, physicians, etc.—should be trained on how to provide care to these individuals [21].

Regarding lactation, the qualitative study conducted by MacDonald et al. described the experiences of trans* men with chestfeeding and the expressions that health professionals used regularly. The authors observed the power of language, the appropriate use of pronouns according to gender, and how words such as “her,” “mother,” “mum,” “breasts,” or “chestfeeding” were annoying and inappropriate. They even observed how touching patients’ chests without their consent caused intense anguish. In that study, the trans* individuals themselves explained and gave guidelines that should be followed in order to make the healthcare provided trans*-competent [4].

### 3.4. Recommendations on Perinatal Cares, Breastfeeding, and Chestfeeding

When support is provided to trans* individuals so that they are able to chestfeed after having undergone chest masculinization surgery, it is essential to explain that chestfeeding does not only consist of giving milk (any amount of milk is important, even if only a few drops). More importance should be given to establishing a link between the parent and the baby. Chestfeeding is a challenge for these men because sucking becomes difficult for the babies due to the lack of tissue and skin; however, a supplement and the “sandwich technique” can be used to shape the chest [13]. It is worth mentioning that sucking is especially difficult if the individuals have undergone the double-incision technique [4].

Trans* men who have not undergone chest masculinization and choose to chestfeed may occasionally wear a chest bandage to handle their dysphoria once milk production is regular and provided that no pressure is exerted on a specific part of the chest. Nevertheless, these individuals should be well informed about the possible risks of the bandage [13]. It is common that these men experience congestion and signs of mastitis, especially after wearing bandages for many years, because there may be glandular tissue involvement [4,13]. The chestfeeding position is essential, because, for example, in a reclined position, the breast tissue stretches, thus making sucking difficult for the baby. The “rugby hold” and the “crossover hold” are the recommended chestfeeding positions [13].

The parents who have chosen not to breastfeed require special support to reduce milk production quickly and safely. The new parents should know that during the immediate postpartum period there will be certain amounts of milk, regardless of having stimulated its production or not. If the individuals do not want to breastfeed or chestfeed, it is recommended to extract only an amount of milk to feel comfortable, and to reduce pain and inflammation by using cold compresses and cold leaves of collard greens [13].

Wolfe-Roubatis and Spatz prepared guidelines for nursing care provided to trans* men in the postpartum period. These guidelines include: (a) health professionals should pay special attention to the language used; (b) if mistakes are made, the health professionals should apologize and correct those mistakes; (c) patients should be asked about the way they want their body parts to be referred to; (d) health professionals should ask patients’ consent to touch their chests; and (e) health professionals should be updated on available resources. In addition, these authors also mentioned aspects to be avoided, such as: (a) asking questions not directly related to the postpartum situation, i.e., future plans, hormonalization, parenting expectations, etc.; (b) presupposing the identity and gender of the individuals; (c) increasing the number of professionals in the consultations, because patients may be induced to perceive themselves as morbid situations. This way, health professionals will be able to focus on the needs of the patients as new parents who need support for breastfeeding or chestfeeding, not for their appearance as trans* individuals [17].

The trans* parent should be encouraged to attend meetings of breastfeeding support groups in a safe and positive environment, because trans* parents may experience feelings of isolation and loneliness. Experiences of trans* families with sensitized lactation consultants can improve group experiences with other families by favoring inclusion and normalization [13].

## 4. Discussion

Gestation and lactation in trans* people is a multifactorial process. The complexity of this fact has been discussed previously in other works [3].

In June 2018, the World Health Organization (WHO) and the APA, after assessing the stigmatizing effect of their diagnostic labels for the classification of transsexuality, considered making a revision of their manuals. In this way, the WHO, in the recent ICD-11, which will enter into force in January 2022, has changed the diagnosis “Transsexualism” to “Gender incongruence of adolescence and adulthood,” considering it as a condition related to sexual behavior [54].

Even according to the APA, despite the attempt to reduce the stigmatizing effect by changing its diagnostic label to the current “Gender dysphoria” in its DSM-V, transsexuality continues to be considered a mental pathology [7].

Under this pathologizing framework, it is not surprising that in most of the consulted bibliography, the barriers that trans* individuals encountered in society also existed in health services. Thus, this fact includes stigmatization, discrimination, invisibility, vexatious and degrading treatment, and inequity in the care provided by health care professionals and institutions [3,5,10,17,25,26].

Many studies have indicated that due to heteronormativity, trans* individuals are excluded from parental and maternal models [35,55,56]. Therefore, trans* parents have a unique position that challenges the binary social construction [57] in which health systems have inherently assigned the fertilizing body and lactation to the female gender [58].

Although scientific production related to transgenderism has been increasing in recent years, studies that specifically address pregnancy and lactation in trans* men are scarce [4,8,10,13,14,17,18,21,47]. The first qualitative study addressing breastfeeding in trans* men was published in 2016 [4]. It is even more striking that we only found one publication addressing breastfeeding in trans* women in the literature [23], a 2018 case report.

Hormone therapy has great impact among trans* individuals. In the induction and establishment of breastfeeding, this impact is even greater in the case of trans* women, given that, in order to be able to breastfeed, they need to complement estrogen therapy with other drugs, such as: spironolactone [1,5], cyproterone acetate, and GnRH agonists [5], whereas trans* men do not require supplementation with medication because with the normal course of pregnancy they already secrete the rest of hormones in a physiological way, thus normally developing breast tissue. The majority of breastfeeding induction protocols for trans* women follow the guidelines of the Newman–Golfarb induction protocol for adoptive mothers or those with children born through surrogacy [42].

There are many publications in the medical literature addressing how to help future mothers in their effort to induce breastfeeding [39,59,60,61]. Although the induction of breastfeeding without prior pregnancy and its permanence in time is possible, it is a long process that requires effort and medical support and supervision [61]. In the case of trans* men, the decision on the alternatives of chest masculinization surgery is of vital importance, because the possibility of chestfeeding in the future—for which the maintenance of glandular tissue is necessary—should be taken into consideration. In this sense, periareolar breast surgery offers better outcomes. [13]. The surgeons who perform these interventions should be updated to offer adequate advice on the choice and indication of surgical techniques, and subsequent follow-up.

Regarding lactation in trans* individuals, we only found two articles that made special mention of nursing care, one related to the care and support for chestfeeding [17] and the other related to perinatal care [62]. Nursing research that addresses transgenderism is developing and trying to improve the care provided, continuously seeking ways to improve techniques and practices. However, it should not be forgotten that counselling is of paramount importance [9]. In this regard, nurses need more training on how to support trans* patients during pregnancy and lactation [17].

As evidenced in the literature, health science schools have been devoting little time to training on health issues related to trans* individuals [27,50,63]. The WHO considers the training of health professionals especially necessary regarding the human rights of trans* individuals, such as the right to dignity, privacy, autonomy, physical and psychological integrity, and the prevention of gender violence [64], so that the barriers that are perceived and reported by these individuals are overcome.

The lack of studies concerning breastfeeding in trans* people is important. This fact necessitates a line of future research, with sufficient evidence studies, to draw scientific conclusions.

## 5. Conclusions

The present study highlights the need of conducting studies in health sciences, especially relating to the care and advice provided by nurses to trans* individuals who are pregnant and want to chestfeed or breastfeed or, both men and women respectively.

The stigma and social discrimination suffered by trans* men when they become pregnant and chestfeed has increased. This stigma and rejection is no different in the health field, which is a barrier to healthcare. Nurses and other health professionals should have special training to enhance their awareness of how this situation should be addressed. They should ensure the experience is perceived as something common and habitual in order to avoid situations in which the individuals hide themselves from others, disguise their chests, or use compressive bandages, which carry specific health risks.

Standardized breastfeeding induction protocols with important hormonal and pharmacological treatments are used in trans* women so that they can breastfeed. These protocols do not always meet expectations, but they produce positive outcomes. In this case, unlike trans* men, feelings of dysphoria are not generated. There is confirmation of gender identity because, socially and culturally, chestfeeding is an inherent fact of the female gender.

Many of the difficulties that trans* individuals face with respect to access to healthcare and the care received result from the culture of heteronormativity, binarism, and predominant cisexuality. There is a lack of training in current health sciences curricula in this regard. If health professionals are trained on sexual and gender diversity, the use of inclusive language, and the multiple possibilities of pregnancy and lactation among trans* individuals, healthcare would be improved and adequate treatments would be prescribed.

## Figures and Tables

**Figure 1 ijerph-17-00044-f001:**
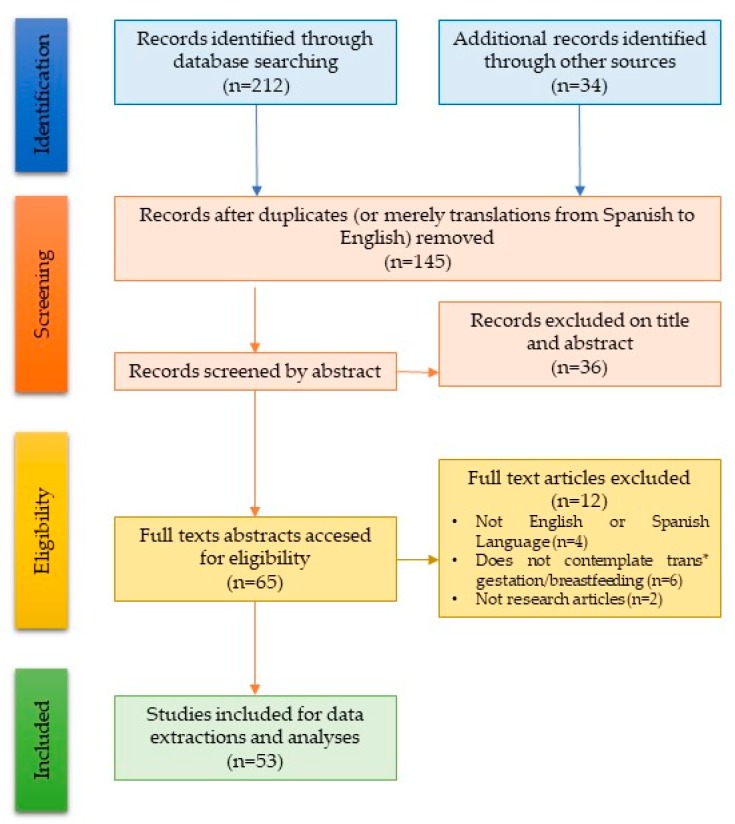
A flowchart showing phases of the literature search for extraction of the most specific literature for the review.

**Figure 2 ijerph-17-00044-f002:**
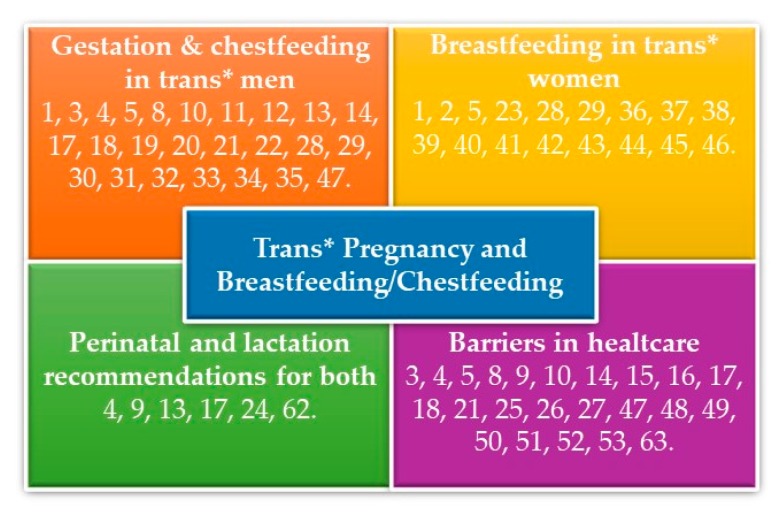
The analysis of the data revealed four broad themes.

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
