# Peer review of "Trans* Pregnancy and Lactation: A Literature Review from a Nursing Perspective"

_ijerph, 2019, doi:10.3390/ijerph17010044_

Round 1

Reviewer 1 Report

General suggestions: Use cisfemale or cismale when discussing cis individuals to prevent confusion. The term trans* may need to be defined for the readership - the fact that the * symbol is meant to be inclusive of all genders. 

Line 20-21: rephrase the sentence. It is difficult to follow as it is currently written. 

Line 24: consider alternate term for bibliography

Line 45: clarify the word 'women' and 'men' - trans* or cis

Line 47: consider alternate word than "perform" hormonal therapies. Consider removing the word perform and just leave it as hormonal therapies

Line 48: word 'transit' may not be necessary 

Line 49: consider rewording "identity-felt" 

Line 69: consider removing word 'tit'

Line 91-92: misleading to say that there were 446 studies. After comparing this written paragraph to the attached diagram of studies it seemed inappropriate to make that statement. Consider rewording to more accurately reflect the search findings. 

Line 111: trans or ciswomen (citation 8)

Line 114: need citation to verify that the majority of transmen choose chest masculinization - if this was in one study or a few that needs to be clarified instead of making a broad statement

Line 120-121: citation on mental health statement

Line 134: not all effects of testosterone are reversible so the phrase "reverse the main changes already established" is not accurate. Please add clarification to this statement

Line 136 - 137: "It has a great damaging impact" is a very strong statement. Verify that it should read "It can have a great damaging impact"

Line 321, 297: Consider using an alternative term and avoiding the use of "transsexuality"

Line 162: consider adding "oestrogen or estrogen" - further in the paper the term 'estrogen' is used so adding this term to line 162 will help it flow 

Line 178: provide additional background on LenoreGoldfarb or clarify if they are a trans*woman

Line 198: consider 'trans*community' or alternate term than trans*group

Author Response

Author's Reply to the Review Report (Reviewer 1)

General suggestions: Use cisfemale or cismale when discussing cis individuals to prevent confusion. The term trans* may need to be defined for the readership - the fact that the * symbol is meant to be inclusive of all genders.

Corrected on lines 47, 57, 114 and 131, in the other paragraphs there is contextualized.

Added a definition of the term trans* in the introduction.

Line 20-21: rephrase the sentence. It is difficult to follow as it is currently written.

Changed. “Pregnancy and lactation involve two aspects that are socially and culturally associated with women. However, there are a few biological differences between male and female breast tissue. Lactation and pregnancy are viable processes that do not depend on sex. Even for the latter, it is only necessary to have an organ capable of gestation”

Line 24: consider alternate term for bibliography

Corrected. Literature

Line 45: clarify the word 'women' and 'men' - trans* or cis

Corrected.

Line 47: consider alternate word than "perform" hormonal therapies. Consider removing the word perform and just leave it as hormonal therapies

Corrected.

Line 48: word 'transit' may not be necessary

Corrected.

Line 49: consider rewording "identity-felt"

Corrected. gender identity that the person feels

Line 69: consider removing word 'tit'

Corrected.

Line 91-92: misleading to say that there were 446 studies. After comparing this written paragraph to the attached diagram of studies it seemed inappropriate to make that statement. Consider rewording to more accurately reflect the search findings.

Right. There was an error in the transcription of data, and they crossed paths with the search results of another article. Figure 1 and 2 are corrected now.

Line 111: trans or ciswomen (citation 8)

Corrected.

Line 114: need citation to verify that the majority of transmen choose chest masculinization - if this was in one study or a few that needs to be clarified instead of making a broad statement

Fixed, added study by Trevor MacDonald.

Line 120-121: citation on mental health statement

Added citation of American Psychiatric Association.

Line 134: not all effects of testosterone are reversible so the phrase "reverse the main changes already established" is not accurate. Please add clarification to this statement

Corrected.

Line 136 - 137: "It has a great damaging impact" is a very strong statement. Verify that it should read "It can have a great damaging impact"

Corrected.

Line 321, 297: Consider using an alternative term and avoiding the use of "transsexuality"

Corrected. The term now used is transgenderism.

Line 162: consider adding "oestrogen or estrogen" - further in the paper the term 'estrogen' is used so adding this term to line 162 will help it flow

All oestrogen terms have been replaced with estrogen

Line 178: provide additional background on LenoreGoldfarb or clarify if they are a trans*woman

Background does not exist; Lenore Goldfarb was the first ciswoman to breastfeed her child born by subrogated gestation after stimulating breastfeeding. Therefore, this protocol was name: “Newman-Goldfarb"   because of   pediatrician Jack Newman who performed it and Lenore Goldfarb, the first ciswoman. It was clarified in the text that she is a ciswoman.

Line 198: consider 'trans*community' or alternate term than trans*group

Corrected.

Reviewer 2 Report

The authors provide a clear summary of previous literature on breast/chest feeding and trans people. Some minor points require revision to ensure that the paper itself is not cisgenderist, but otherwise a useful contribution to the literature. 

1) The definition of trans people is itself cisgenderist. More appropriate would be 'trans people experience their gender as different to that normatively expected of their assigned sex'. 'Correspond' itself is cisgenderist (ie it presumes sex determines gender)

2) The word 'inconsistency' is also a form of cisgenderism (sentence 3)

3) Phrases like 'lack a female reproductive system' is itself cisgenderist. Better to say 'do not have a reproductive system that allows for gestation'

4) 3.2 (and also the introduction) should introduce the term 'chest feeding' as an alternative to 'breast feeding'. The authors also need a little more nuanced account of chest feeding for trans men: not all such men find it dysphoria inducing

5) The use of the asterisk after trans is not widely accepted as unproblematic. The authors need to explain their usage (particularly given they are talking about men and women, so the purportedly inclusive asterisk is less relevant for this paper).

6) 3.4 would be better if the language 'breastfeeding/chestfeeding' is used, so as not to contribute to the marginalisation of trans men

7) Is it due to heteronormativity or cisgenderism that trans people are excluded from parental models?

8) This statement is entirely untrue: "There are few publications in the medical literature addressing how to help future mothers in their effort to induce breastfeeding". There is endless literature on cis mothers and breast-feeding. If you mean trans people then say that. 

9) The abstract needs rewriting to reduce cisgenderism. People dont need a female reproductive organ to reproduce, they need to be capable of gestation (indeed, some cis women with an internal reproductive system cannot gestate). 

Author Response

Author's Reply to the Review Report (Reviewer 2)

1) The definition of trans people is itself cisgenderist. More appropriate would be 'trans people experience their gender as different to that normatively expected of their assigned sex'. 'Correspond' itself is cisgenderist (ie it presumes sex determines gender)

Corrected in introduction.

2) The word 'inconsistency' is also a form of cisgenderism (sentence 3)

The term has been changed to discrepancy.

3) Phrases like 'lack a female reproductive system' is itself cisgenderist. Better to say 'do not have a reproductive system that allows for gestation'

Thanks for the appreciation, that's right. Fixed.

4) 3.2 (and also the introduction) should introduce the term 'chest feeding' as an alternative to 'breast feeding'. The authors also need a little more nuanced account of chest feeding for trans men: not all such men find it dysphoria inducing

In the context of trans* men, the breastfeeding term has been changed to chestfeeding. In the case of trans* women has been left, since this is what the bibliography refers to.

5) The use of the asterisk after trans is not widely accepted as unproblematic. The authors need to explain their usage (particularly given they are talking about men and women, so the purportedly inclusive asterisk is less relevant for this paper).

According to a point of other reviewer, a clarification has been made in the introduction of the symbol *. We believe that it is necessary to be inclusive with all identities/genders as there are non-binary people who can also feel identified with the text.

6) 3.4 would be better if the language 'breastfeeding/chestfeeding' is used, so as not to contribute to the marginalisation of trans men

Related to trans* men has changed the term breastfeeding to chestfeeding throughout the text.

7) Is it due to heteronormativity or cisgenderism that trans people are excluded from parental models?

Both are expected to be the ‘ideal’ prototype of cisgender and heterosexual couples. There is a presumption of cisgender and heterosexuality in society and that creates an ‘ideal’ parental model, in the perspective of society.

8) This statement is entirely untrue: "There are few publications in the medical literature addressing how to help future mothers in their effort to induce breastfeeding". There is endless literature on cis mothers and breast-feeding. If you mean trans people then say that.

Right, it's a mistake. Fixed.

9) The abstract needs rewriting to reduce cisgenderism. People dont need a female reproductive organ to reproduce, they need to be capable of gestation (indeed, some cis women with an internal reproductive system cannot gestate).

Thanks. You’re right. Here is the new redaction

“Pregnancy and lactation involve two aspects that are socially and culturally associated with women. However, there are a few biological differences between male and female breast tissue. Lactation and pregnancy are viable processes that do not depend on sex. Even for the latter, it is only necessary to have an organ capable of gestation. Ways to favour mammogenesis and lactogenesis in trans* women have been established. There are protocols to promote lactation in trans* women, usually used for adoptive mothers or those whose children have been born through gestational surrogacy. Chestfeeding a baby could be the cause of feelings as diverse as gender dysphoria in the case of trans* men, and euphoria and affirmation of femininity in trans* women. This study involves a review of the available scientific literature addressing medical aspects related to pregnancy and lactation in trans* individuals, giving special attention to nursing care during perinatal care. There are scarce studies addressing care and specific nursing care in trans* pregnancy and lactation.  Our study indicates the factors that can be modified and the recommendations for optimizing the care provided to these individuals in order to promote and maintain the lactation period, in search of improvement and satisfaction with the whole process”